# Adolescents Accessing School-Based versus Family Planning Clinics: Chlamydia and Gonorrhea Testing and Treatment Outcomes

**DOI:** 10.3390/biology11040521

**Published:** 2022-03-29

**Authors:** Meghna Raphael, Allyssa A. Abacan, Peggy B. Smith, Mariam R. Chacko

**Affiliations:** 1Department of Pediatrics, Baylor College of Medicine, One Baylor Plaza, Houston, TX 77030, USA; mchacko@bcm.edu; 2Department of Obstetrics & Gynecology, Baylor College of Medicine, One Baylor Plaza, Houston, TX 77030, USA; abacan@bcm.edu (A.A.A.); peggys@bcm.edu (P.B.S.)

**Keywords:** school-based clinics, family planning clinics, sexually transmitted infections, STI screening in adolescents, chlamydia and gonorrhea testing

## Abstract

**Simple Summary:**

Sexually Transmitted Infections, such as gonorrhea and chlamydia, are common in school-aged youths. These infections can cause significant health problems if not identified and treated early. Current estimates of how widespread these infections are among students receiving healthcare in schools, the role school-based clinics may have in gonorrhea and chlamydia testing and treatment, and information on how the COVID-19 pandemic affected testing and treatment, are lacking. This study was conducted in an urban metropolitan area in the United States, and included 2439 patients aged 13–17 years over a 2-year period. The patients were seen at four school-based and five family planning clinics. We found that 35% of those tested in school-based clinics were positive for chlamydia, and 10% were positive for gonorrhea. The rates of these infections were higher than previous reports from school settings (both pre-COVID-19 and the during COVID-19 pandemic). School-based clinics were able to treat patients with gonorrhea and chlamydia much faster (average ~6 days) than the family planning clinics (average ~18 days). This study shows us the critical role that school-based clinics play in the testing and treatment for gonorrhea and chlamydia infections, and the value of strengthening the services these clinics provide.

**Abstract:**

The prevalence and treatment of chlamydia (CT) and gonorrhea (GC) at school-based clinics (SBCs) requires revisiting. To assess whether clinic type influences CT/GC testing and treatment for minors (individuals 13–17 years of age), our study compared four SBCs with five family planning clinics (FPCs) in the Houston, Harris County metropolitan area of Texas, USA for: (1) the prevalence of CT/GC infection (pre-COVID-19 and during COVID-19); (2) treatment rates at the last positive diagnosis; and (3) the time, in days, from testing-to-diagnosis and testing-to-treatment. Between January 2019 and December 2020, 2439 unique patients (1579 at SBCs, 860 at FPCs) were seen. Of the 1924 tests obtained, 39.2% and 15.9% were positive for CT and GC, respectively. The prevalence of CT and GC at SBCs was similar prior to COVID-19 vs. during the COVID-19 pandemic. SBCs were able to provide treatment significantly faster after diagnosis (mean, 6.07 days; 95% CI, 3.22–8.90; 94.7% were within 30 days) than FPCs (mean, 17.60 days; 95% CI, 10.15–25.12; 84.7% were within 30 days) (*p =* 0.0257). This comparison within our large clinic system, with consistent clinical management protocols, suggests that SBC care may be critical to ensuring optimal sexually transmitted infection management in minors.

## 1. Introduction

In 2019, chlamydia (CT) and gonorrhea (GC) infections were reported at an all-time high in the United States. CT infection rates in 15–19-year-old females and males were at 3333.8/100,000 and 1009/100,000, respectively. In the same year, CT infection rates in the Houston, Harris County area for all age groups ranked fourth in the country, at 504.3/100,000, and GC infection rates ranked sixth in the country, at 328.6/100,000. Age-specific data for the area are currently unavailable [1]. In order to decrease the rate of medical complications, the United States Public Service Task Force (USPSTF) and the Centers for Disease Control (CDC) recommend annual CT and GC screening in sexually active women under 25 years of age [2,3]. While annual screening in all males is not recommended, in high-prevalence clinical settings, such as adolescent clinics and correctional facilities, and upon testing positive for a sexually transmitted infection (STI), males who have sex with females should be screened annually. Males that have sex with males should also receive annual screening, with more frequent risk-based screening (every 3–6 months) on a case-by-case basis [3]. With the widespread availability of urine nucleic acid amplification testing, these screenings can be performed in an easy and noninvasive manner.

School-based clinic (SBC) sites may be convenient locations for increasing the access to appropriate screening among adolescents [4]. The CDC recommends that schools increase access to broad sexual health services, including risk assessments, STI surveillance through testing, and STI treatment and education [5]. Despite these recommendations, access to sexual health services amongst students is limited. Available prevalence estimates from studies conducted at SBCs/health centers between 1998–2010 showed rates of CT to be 4.75–19%, with differences noted between sex and geographic location [6,7,8,9]. Schoolwide population-based studies, with interventions promoting screening, have shown prevalence rates of incidental CT in females ranging from 5.6% (California) to 13% (New Orleans) [9,10,11,12,13]. However, these available estimates of CT/GC infection rates do not reflect trends from the last decade, and are unavailable for other geographic areas. With the significant changes in the overall rates of STIs in the general population over last decade, it is likely that the previously estimated prevalence rates are no longer reflective of the STI rates reported at SBCs. Furthermore, there is a need to assess treatment rates among adolescents that test positive for CT/GC infection.

Once an infection is detected through screening, timely treatment is critical. When CT and GC infections are left untreated in females, they can cause complications, such as pelvic inflammatory disease (PID), ectopic pregnancy, chronic abdominal pain, and infertility. Similarly, in males, untreated CT and GC infections can cause epididymitis and, rarely, disseminated infection [2]. Treatment completion rates vary widely based on the setting. Previous studies have demonstrated that ~25% of patients that tested positive for CT and GC, and that were seen at public health department clinics or emergency departments, remained untreated [14,15]. Treatment rates among minors in juvenile detention facilities has been noted to be significantly lower, with >50% of those detected to have an STI remaining untreated upon release from the detention center [16]. Among adolescent females seen at an urban health center, ~10% were untreated after 30 days [17]. Barriers to treatment, as reported by adolescents, include lack of transportation, insufficient time, and poor knowledge of STI consequences if left untreated [18]. These studies demonstrate that the location of care may be a key factor to ensuring successful treatment. Direct comparisons of treatment outcomes between SBC sites and other teen health service providers (such as family planning clinics, FPCs) are warranted.

An additional consideration is the impact of the COVID-19 pandemic on the screening, detection, and treatment of CT and GC. *Disruption* to STI services occurred during the COVID-19 pandemic due to multiple factors, including the deferment of routine screening, inadequate expansion of telehealth options in some settings, diversion of personnel, and medication shortages. During the COVID-19 pandemic in 2020, reported CT and GC cases in the U.S. dropped to 50% of 2019 levels for CT, and 71% for GC. By December 2020, the reported cases resurged to 101% of pre-pandemic levels for CT, and 135% for GC [19]. The impact of the pandemic on SBC STI services has not been studied.

Thus, this study addresses important gaps in the literature, including: the lack of current CT and GC prevalence estimates in SBC settings, the impact of the COVID-19 pandemic on SBC STI services, and the treatment outcomes for adolescent minors with CT/GC at SBCs vs. FPCs in a previously unreported geographic area. The purpose of this study was to describe and compare SBCs with FPCs for: (1) the prevalence of CT and GC infection (pre-COVID-19 and during COVID-19); (2) treatment rates; and (3) the time, in days, from testing-to-diagnosis and testing-to-treatment for infected individuals, with a subgroup analysis separating those that were treated presumptively, among adolescent minors that sought care at our nine clinic locations between 1 January 2019 to 31 December 2020.

## 2. Methods

### 2.1. Setting

This retrospective study was conducted in a nine clinic-system that provides primary preventive and reproductive health care to adolescents and young adult (AYA) males and females, primarily 13 to 24 years of age, in the Houston, Harris County metropolitan area of Texas, USA. The institutional review board (IRB) of Baylor College of Medicine and its affiliated hospitals provided approval for the study. The study underwent expedited review, and was classified as “research not involving greater than minimum risk”. A waiver of consent was requested and approved by the IRB. The clinic system is predominantly state-funded for family planning health programs, and has been an important safety net for Medicaid, low-income, and uninsured AYA individuals, conducting over 20,000 visits a year. Of the nine clinics, four are SBCs (three public high schools and one charter school) and five are FPCs. The four SBCs are located directly on the high schools’ campus, and are open to both students and community members that fall within the age range (school-linked). The SBCs receive the support of the independent school district and school administrators. Preventive visits include immunizations, wellness exams, pregnancy tests, hormonal and non-hormonal birth control methods, and STI screening and treatment. For successful implementation of SBC STI screening, diagnostic testing, and treatment programs in AYAs, self-consent and confidential care are critical. In Texas, minors (≤17 years) are permitted by state law to self-consent to STI testing and treatment (Texas Family Code §§32.003, 2013) [20]. This also allows our clinics to provide confidential STI testing (including GC and CT) and treatment services as recommended by the CDC and the School-Based Health Alliance [21,22].

The clinics follow the CDC STI guidelines for testing and treatment. The guidelines include testing all sexually active adolescents, with their consent, regardless of symptoms [3]. All clinics follow the same policies and procedures for the intake of sexual and STI history, physical examination, diagnosis, follow up of patients with positive STI test results, and treatment. CT and GC specimens (urine, vagina, cervix, anal, and/or oral sites) are obtained for testing and sent out to a laboratory contracted to state-funded family planning and reproductive health clinics. The Roche Cobas^®^ assay, a PCR-based nucleic acid test, is used for the qualitative detection of CT and GC. Presumptive treatment is provided for female patients with symptoms of cervicitis or PID, for male patients with urethritis, and for patients with an established exposure to CT or GC [3]. Three attempts are made to contact a patient with a positive result. The first attempt occurs as soon as possible and within seven days of receiving the test result. If the patient has not been treated, the second and third attempts are each made seven days from the prior contact date. If a patient has not returned for treatment after 30 days, the case is referred to the local health department for follow up.

In 2020, the COVID-19 pandemic impacted the SBC’s ability to operate as normal. From mid-March through to May 2020, the three public high school-based clinics were closed; thus, these clinics did not have the ability to see any patients, students or community members, for any health care services, including STI testing and treatment, during that time period. All patients were directed to one of the five open FPC locations or the one charter school location. The public high school-based clinics reopened in June 2020; however, summer school was not in session. The independent school district opened back up for the 2020–2021 academic year in August, with a hybrid set-up. One-third of the students attended in-person, while the other two-thirds of the student population opted to attend virtually.

### 2.2. Study Population

All patients <18 years of age that sought care at any one of our nine locations between 1 January 2019 to 31 December 2020, were included in the study (Figure 1. Study Population). Data were extracted from clinical billing software and electronic medical records (EMR).

### 2.3. Study Measures

Variables abstracted were age, race, sex, clinic location type (SBC/FPC), CT/GC testing (yes/no), CT/GC diagnosis (positive/negative), and CT/GC treatment (yes/no), as well as dates of CT/GC testing, results, and treatment (in days). CT/GC testing represented diagnostic testing in patients with symptoms and the screening of asymptomatic patients. Individuals with zero days from screening-to-treatment were categorized as having been treated presumptively.

### 2.4. Preparation of Study Database

A visual depiction on the preparation of the study database can be seen in Figure 2. 

(1)Demographic variables and testing dates were extracted from a billing software. All patient encounter dates and billed services were generated using the billing software. CT/CG results are uploaded to the billing software, which are then automatically sent to the patient’s chart in the EMR for clinical provider review.(2)A CT/GC testing, diagnosis, and treatment list from 1 January 2019 to 31 December 2020 was created by obtaining CT/GC diagnosis, treatment, and dates from the EMR using ICD-10 codes (A54 = Gonococcal Infection and A56 = Other Sexually Transmitted Chlamydial Diseases). Treatment was confirmed by medication prescriptions documented in the medical visit by the clinical provider.(3)A master database was then created to compile all variables together and match each patient visit with each variable of interest. The master database included all patient visits from 1 January 2019 to 31 December 2020, and included the patient demographics. The CT/GC testing, diagnosis, and treatment lists were used to match the master database with the respective patient and date of visit. The specific diagnosis date was matched to the correct testing date, and the treatment was matched to the correct diagnosis date. The master database was then filtered to only include adolescent minors.(4)Outliers were noted when data were retrieved from the billing system and EMR. These included missing data upon treatment completion, and/or prolonged time to treatment completion. This was likely due to EMR data entry errors leading to incomplete data generation during data extraction. Hence, the study investigators conducted a targeted chart review to manually validate outlier data points (i.e., obtain missing time data if the extracted data indicated non-treatment, or time-to-diagnosis/treatment listed at >10 days, determined from clinical experience).(5)The number of days from testing-to-diagnosis, diagnosis-to-treatment, and testing-to-treatment for the last positive diagnosis date was determined. This approach represented at least one positive test per patient across sites. Obtaining this information, we were able to study time outcomes for 402 tests (134 at SBCs vs. 268 at FPCs) of the 1061 positive tests (40.2% of positive tests at SBCs and 36.8% positive tests at FPCs).

### 2.5. Analysis

All analyses were conducted using STATA version 17.0. This retrospective chart review study sought to estimate: (1) the prevalence of CT and GC infection, as assessed by dividing total number positive tests by the total of tests obtained (pre-COVID-19 and during the COVID-19 pandemic); (2) treatment rates for the last positive diagnosis, and; (3) time, in days, from testing-to-diagnosis and testing-to-treatment of a positive CT and GC test for the last positive diagnosis. A subgroup analysis of cases excluding individuals that were treated presumptively was conducted. Frequencies were calculated to generate descriptive statistics and cross-analyses of the demographic variables. Chi-square analyses were performed to compare demographics, CT/GC prevalence, treatment rates, and treatment within 30 days between SBC and FPC sites. In addition, *t*-tests were performed to analyze continuous variables, including time from testing-to-diagnosis, diagnosis-to-treatment, and testing-to-treatment, to make comparisons between SBC and FPC sites (including and excluding those treated presumptively). Significance was set at *p* = 0.05.

## 3. Results

Two-thousand four hundred and thirty-nine unique patients <18 years of age were seen between January 2019 and December 2020. The number of adolescents seen at the SBCs was higher than at the FPCs (1579 and 860 respectively) (Table 1). Differences were noted in age, race, and sex. Patients seen at the SBCs were marginally younger (15.9 years, range 12–17 years vs. 16.02 years, range 13–17 years) and were more likely to be male (58.5% vs. 13.8%) (*p* < 0.0001), as compared to those seen at the FPCs. Significant differences were noted between race, with the majority of patients self-identifying as Black in FPCs and White in SBCs (*p* < 0.0001). Of note, 93.3% of all patients were of Hispanic ethnicity in 2019–2020.

### 3.1. Frequency of Patients Tested and Number of Tests Obtained

Over the course of two years, of the 2439 unique patients, 1249 (51.2%) were tested for CT/GC. Ninety percent of patients at FPCs (774 of 860) received at least one test for CT/GC vs. 30.1% (475 of 1579) of patients at SBCs. These 1249 patients had a total of 1924 CT/GC tests, the majority of which were performed at FPCs (1180 of 1924 tests vs. 744 of 1924 tests at SBCs).

### 3.2. Prevalence of Positive Results

Of the 1924 total tests performed, 39.2% (755) were positive for CT and 15.9% (306) were positive for GC. The prevalence of both CT and GC was significantly higher in the FPCs compared to SBCs; CT prevalence was 34.9% at SBCs (260 of 744 tests) and 41.9% at FPC (495 of 1180 tests) (*p =* 0.002); GC prevalence was 9.8% at SBCs (73 of 744 tests) and 19.7% at FPCs (233 of 1180 tests) (*p* < 0.0001). Significant differences in testing and diagnosis rates were noted between sex. Male patients that received testing were more likely to have at least one positive test result; 85 of 209 (41.1%) of males tested vs. 317 of 1040 (30.5%) of females tested, *p =* 0.003.

### 3.3. Impact of COVID-19 Pandemic on Testing Rates and Prevalence Estimates

The impact of the COVID-19 pandemic on routine screening was significant. Based on the testing rates between January 2019 and March 2020 (1357 tests), the clinics performed 30% fewer tests between April 2020 and December 2020 than would have been anticipated in this 9-month period (567 tests conducted, projected ~814). This effect was more significant at the SBCs, which performed 44% fewer tests after the onset of the COVID-19 pandemic (187 tests conducted, projected ~334). Test positivity rates for CT and GC remained similar prior to vs. during the COVID-19 pandemic at school-based sites (for CT, pre-COVID-19 = 33.7%, during COVID-19 = 38.5%, *p =* 0.238; for GC, pre-COVID-19 = 10.4%, during COVID-19 = 8.02%, *p* = 0.341). However, at the FPCs, the test positivity rates for both CT and GC were significantly higher during the COVID-19 pandemic as compared to the 16 months prior (CT rate pre-COVID-19 = 38.1%, during COVID-19 = 50%, *p* < 0.0001; GC rate pre-COVID-19 = 16.3%, during COVID-19 = 26.8%, *p* < 0.0001).

### 3.4. Treatment Outcomes

Based on the location of care, significant differences were noted in the proportion of patients that tested positive at their last CT/GC test. The proportion of positive patients that completed treatment (including those that were treated presumptively) at their last positive diagnosis was statistically higher at SBCs vs. FPCs (98.5% vs. 92.1%, *p =* 0.01). Treatment completion rates were similar pre-COVID-19 (93.4%, 95% CI = 90.5–96.4%) and during COVID-19 (96.1%, 95% CI = 92.7–99.5%, *p =* 0.284).

Among patients with a positive test, significant differences were noted in the time from testing-to-diagnosis, diagnosis-to-treatment, and testing-to-treatment for the last positive diagnosis date, based on location of care. SBCs were able to provide treatment much faster after testing (mean 6.07 days, 95% CI = 3.22–8.90; 94.7% were within 30 days) than FPCs (mean 17.60 days, 95% CI = 10.15–25.12; 84.7% were within 30 days) (*p =* 0.0299). These findings are presented in Table 2.

Of the 402 total patients that received treatment for a positive test, 17.9% (55.2% of males and 7.9% of females) received treatment presumptively on the day of testing based on symptoms or exposure to CT or GC. The proportion of patients that received presumptive treatment did not differ based on location of care (22.4% at SBCs vs. 15.7% at FPCs, *p =* 0.098). Since patients that are treated presumptively may not have experienced the same barriers to return to care for treatment, we analyzed the time-to-treatment data for the subgroup of patients that tested positive but did not receive presumptive treatment. Among patients with a positive test that were not treated presumptively, significant differences were noted in the treatment received at last diagnosis, time from testing-to-treatment, and time from testing-to-diagnosis, based on location of care. Overall, SBCs were able to provide treatment significantly faster after testing among patients that did not receive presumptive treatment (Table 3).

## 4. Discussion

Our study was conducted using a nine-clinic system in a previously unstudied geographic location of the United States. We compared the data for patients seen at SBCs vs. FPCs that shared the same STI protocol and procedures. Despite the frequency of testing, and the test positivity rates being lower at SBCs as compared to FPCs, we found a high prevalence of CT and GC cases among the SBCs, both during and prior to the COVID-19 pandemic. At SBCs, the frequency of testing was significantly higher for female patients, though males that were tested were more likely to test positive for an infection. Treatment rates did not vary by gender. Patients seen at SBCs were significantly more likely to complete treatment, and do so much faster, as compared to FPCs.

### 4.1. Prevalence

Prior studies reported estimated prevalence rates of ~35% for CT and ~10% for GC at SBCs [6,7,8,9], which are lower than those reported here. These prior studies at school-based sites found CT rates of 12–19% in females (Baltimore) and 4.75–7.5% in males (Denver, New Orleans, Baltimore, MD, USA). All the previously reported data from SBCs were conducted more than a decade ago, and national trends demonstrating an increase in CT and GC infections, particularly for certain geographic locations (southern USA), may be responsible for these differences [1].

### 4.2. Treatment Rates and Time-to-Treatment

Separating treatment site from the testing site is an important variable influencing treatment completion and time-to-treatment. The majority of patients (~92–98% depending on site) that tested positive completed treatment in our clinics. Our results are similar to the findings from a Chicago schoolwide screening program, which reported that 89.4% of students with positive test results received treatment [4].

The time from testing-to-treatment was, on average, ~11 days less at SBCs as compared to FPCs. Even with the exclusion of patients that were treated presumptively, the time from testing-to-treatment was significantly less at SBCs vs. FPCs. This indicates that same-day treatment was not the reason for the better outcomes at the SBCs. The rates of presumptive treatments did not differ between the SBCs and FPCs in our clinic system. Our rate of presumptive treatment was 17.9%, and was higher than the ~10% rate reported in the literature for FPCs [14]. While presumptive treatment is one approach to decrease the time-to-treatment, this needs to be balanced with improving the accuracy when making a presumptive diagnosis and responsible antibiotic stewardship to minimize the risk of increasing resistant strains [23].

The mechanisms by which SBCs can provide more timely treatment are multifold, most important of which may be the ease of access to care during school hours. This is particularly vital for minors requiring confidential STI services, or to those that may not have access to transportation to offsite clinics [4]. The cost of testing and treatment was not a barrier to seeking care at any of our clinic sites, since patients receive services at no cost. Another factor that can influence the time-to-treatment may be the volume of patients seen at the clinic. In our study, 15.3% of patients were not treated within 30 days at FPCs vs. 5.3% at SBCs. A previous study of adolescents found this to be 17–29.4%, with differences noted based on the volume of patients seen at a clinic (28.6% in high-volume clinics vs. 18.8% in low-volume clinics) [14]. While fewer minors were seen at our FPCs compared to SBCs, our FPCs are higher volume clinics that serve many young adults. This indicates that a greater effort and focus on adolescent minors at the FPCs is needed.

### 4.3. Effects of the COVID-19 Pandemic

During the initial phases of the COVID-19 pandemic in the USA, the CDC recommended that individuals with genital symptoms and/or that have been exposed to an STI, as well as pregnant women with syphilis, should be given priority. Routine screening visits in clinics that remained open were to be deferred until clinic schedules allowed for increased number of patient visits [24]. This led to a ~50–60% decline in sexual health screening nationally. Additionally, there was a similar decrease in the capacity to treat STIs or follow up on positive cases due to the diversion of health department work force to COVID-19 efforts. Additionally, in the summer and fall of 2020, shortages of STI testing kits and azithromycin were announced [25]. Despite the impact of the COVID-19 pandemic on STI services nationwide, our five FPCs, and one charter school clinic, were neighborhood clinics that provided essential services to the community. CDC recommended that COVID-19 safety guidelines were followed, and the six locations were able to remain open throughout the COVID-19 pandemic. They had access to treatment and testing supplies despite shortages reported elsewhere in the country. We also expanded access to telehealth rapidly during this time frame. However, our clinics experienced lower patient volumes between April and December 2020, as stay-at-home orders were in place. The impact was particularly severe at SBCs, as these clinics were subject to closures between April–June 2020. In the months following, while students and community members had access to all our clinics, including SBCs, the students were not on campus for educational instruction, and in-person appointments were limited to allow for social distancing. This led to a decrease in the utilization of clinics, particularly by minors, and less CT and GC testing at SBCs and FPCs. However, test positivity rates remained high at both SBCs and FPCs. This may be due to adolescents seeking care when they were symptomatic or perceived themselves at elevated risk for infection, despite the pandemic. Treatment completion rates were similar during this time frame, as compared to the pre-COVID data.

### 4.4. Strengths and Limitations

The novelties of this study include the updated information on the prevalence of CT/GC at SBCs and FPCs in a large urban setting, data to support the value of SBCs in CT/GC testing and treatment, new information on the effect of the COVID-19 pandemic on STI services for adolescents, and its pragmatic approach to data selection with a large dataset.

The most recent prevalence estimates of CT/GC in school settings are more than a decade old, and include limited geographic areas. Given the changing trends in STI prevalence over the last decade in the general population, updated data for school-based clinics is vital. We found that, in accordance with the national trends, CT and GC rates are significantly higher than previously reported for SBCs. This data also expands our knowledge of STI rates in a large, publicly and state-funded SBC and FPC system in the southwestern US.

Additionally, the effect of the COVID-19 pandemic on SBC STI services was previously not available. Through this study, we found that, while the COVID-19 pandemic led to reduced testing, minors continued to exhibit high rates of CT and GC at both SBCs and FPCs.

The sampling approach used here was effective at identifying differences between sites. Our findings suggest that STI treatment is likely to be completed in a timely manner if minors have easily accessible services, such as at SBCs. Head-to-head comparisons within a large clinic system, such as ours, with established consistent protocols for all sites, suggests that the location of care may be critical for ensuring optimal STI management in minors. This has important policy implications and provides support for the recommendation to strengthen school-based clinics in terms of comprehensive sexual health services.

The limitations of this study include:(1)Our SBCs are open to the public; hence, we cannot clearly distinguish whether minors seen at an SBC were school students or community members.(2)We were unable to retrieve information with regards to partner services, such as expedited partner treatment, an important but challenging component of STI prevention. Similarly, we were unable to investigate the differences in the recommended 3-month testing protocol for reinfection after treatment between SBCs vs. FPCs [3]. Efficient mechanisms to enter and retrieve this information from the EMR are key to study this important quality of care metric.(3)The study is only generalizable to populations within a public urban clinic system within the United States.

### 4.5. Future Directions

Given the high positivity rates, schoolwide population-based screening in our program should be considered [10,11,12]. While designing such programs, it is important to consider individual factors (STI risk, willingness to test and treat), microsystem factors (safe clinical spaces, and trauma-informed adolescent friendly care), community factors (self-consent laws and confidentiality), and national factors (nationwide STI control efforts), among others [26]. Some strategies that have been successfully used to screen large numbers of students in educational settings have included voluntary and opportunistic screening at SBCs, classroom-based screening, screening during sports physicals/health examinations, and screening during campus events [27]. In contrast to clinic-based testing, schoolwide population-based STI testing programs require preparation and a partnership between the health department, school, parents, and clinical leadership [28]. Barriers related to funding streams and the cultural climate in the southern US may need to be addressed. Efforts focused on further studying the quality of services at SBCs, such as expedited partner services, and appropriate follow up services, such as rescreening, are areas requiring future study. Data collection and effective mechanisms to obtain this information from the EMR should be developed. The promotion of SBCs for reproductive health care, such as STI services, are needed, as these clinics increase access to services, increase convenience, and decrease the time-to-treatment. If a minor is seen at a FPC, aggressive efforts to reach patients with positive results, including collecting accurate and alternative contact information, and addressing barriers that may prevent return for treatment, may improve outcomes [29]. Consideration needs to be given to providing point-of-care testing for CT and GC at SBCs and FPCs to increase same-day treatment [30]. The COVID-19 pandemic raised unique challenges due to clinic and school closures, nationwide supply shortages, limited capacity to accommodate social distancing, changes in individuals’ attitudes towards receiving health care, and/or new barriers leading to the decreased use of STI services at SBCs and FPCs. Novel approaches to address these barriers, such as leveraging telehealth, partnering with school administrators and parents to maximize healthcare utilization, and minimizing the disruption to these essential services through workforce and supply chain strengthening at the level of the health department, are vital.

## 5. Conclusions

This study demonstrated the high prevalence of CT/GC in adolescents seeking care at urban SBCs and FPCs, prior to and during the COVID-19 pandemic. SBCs are able to provide consistent and quick treatment to patients detected to have CT/GC, and hence are critical locations for optimal onsite management of minors with STIs.

## Figures and Tables

**Figure 1 biology-11-00521-f001:**
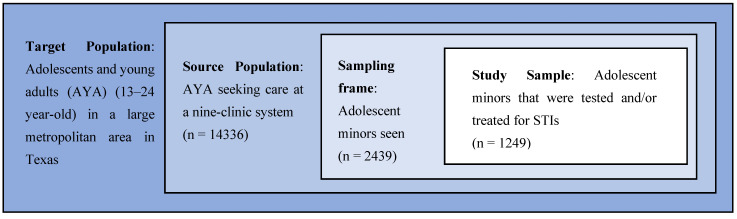
Study Population.

**Figure 2 biology-11-00521-f002:**
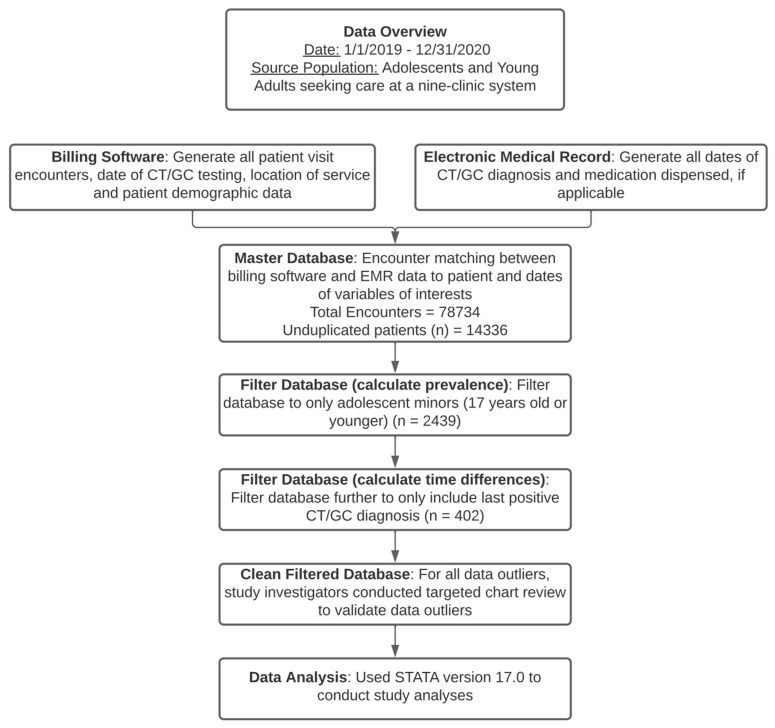
Dataset Preparation.

**Table 1 biology-11-00521-t001:** Demographic characteristics of 2439 unique adolescents seen at school-based (SBC) and family planning clinics (FPC).

	SBC N = 1579	FPC N = 860	*p*-Value
Age Mean (95% CI)	15.9 (15.84–15.95)	16.02 (15.93–16.09)	0.017
**Sex**			
Male	924 (58.5%)	119 (13.8%)	<0.001
Female	655 (41.4%)	741 (86.1%)	
**Race**			
White	1158 (73.3%)	342 (39.8%)	<0.001
Black	382 (24.2%)	510 (59.3%)	
Asian	38 (0.02)	7 (0.09%)	
Other	1	1	

**Table 2 biology-11-00521-t002:** Chlamydia and gonorrhea testing and treatment outcomes at last positive test (including patients treated presumptively) for adolescents at school-based (SBC) and family planning clinics (FPC).

	SBC	FPC	*p*-Value
Patients Tested (N = 1249)	475 of 1579 (30.1%)	774 of 860 (90%)	<0.001
Female (N = 1040)	381 of 924 (41.2%; 38–44.4%)	659 of 741 (88.9%; 86.7–91.1%)	<0.001
Male (N = 209)	94 of 655 (14.4%; 11.7–17.03%)	115 of 119 (96.6%; 93.4–99.8%)	<0.001
Tested positive at last test (N = 402)	134 of 475 (28.2%)	268 of 774 (34.6%)	0.028
Female (N = 317)	101 of 381 (26.5%; 22.1–30.9%)	216 of 659 (32.8%; 29.2–36.4%)	0.034
Male (N = 85)	33 of 94 (35.1%; 25.4–44.7%)	52 of 115 (45.2%; 36.1–54.3%)	0.138
Treatment received at last diagnosis	132 of 134 (98.5%)	247 of 268 (92.1%)	0.010
Testing-to-diagnosis (days)	1.50 (1.21–1.79) *	2.08 (1.85–2.32) *	0.003
Diagnosis-to-treatment (days)	4.71 (1.79–7.62) *	15.50 (8.08–23.00) *	0.044
Testing-to-treatment (days)	6.07 (3.22–8.92) *	17.60 (10.15–25.12) *	0.030

NOTE: * Values are represented as Mean (95% CI).

**Table 3 biology-11-00521-t003:** Chlamydia and gonorrhea treatment outcomes for subgroup of patients that were not treated presumptively.

	SBC N = 104	FPC N = 226	*p*-Value
Treatment received at last diagnosis	102 (98.1%)	205 (90.7%)	0.015
Testing-to-treatment (days)	7.86 (4.23–11.49) *	21.25 (12.3–30.2) *	0.042
Testing-to-diagnosis (days)	1.81 (1.48–2.14) *	2.35 (2.1–2.6) *	0.015
Diagnosis-to-treatment (days)	6.05 (2.39–9.69) *	18.8 (9.94–27.79) *	0.051

NOTE: * Values are represented as mean (95% CI).

## Data Availability

The data presented in this study are available on request from the corresponding author. The data are not publicly available due to data source due to sensitivities and confidentiality of patient health information.

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
