# Peer review of "Adolescents Accessing School-Based versus Family Planning Clinics: Chlamydia and Gonorrhea Testing and Treatment Outcomes"

_biology, 2022, doi:10.3390/biology11040521_

Round 1

Reviewer 1 Report

Abstract

1) Authors are recommended to provide structured-based abstract including background, method, result and conclusion.

Introduction 

1) Authors are recommended to provide references for sentence in lines 35-37

2) Authors are recommended to provide reference for sentence in lines 42-44

3) It would be productive if authors mention to (if there is any) standard protocol or recommendation by WHO or CDC regarding the subject.

Method

Setting

1) Authors are recommended to explain about IRB in the setting at early stage of the study

Result

1) In result section, Fig1.... the number of minor participants mentioned as 2439, however the population in Fig 2 has mentioned as 2394. It would be helpful if authors explain about the difference

Discussion

1) It would be productive if authors conclude from their studies to give some recommendation for low and middle income countries to well-manage screening, diagnosis and early treatment of CG/TC during pandemic. 

Reviewer 2 Report

Dear author's

I am pleased to review your manuscript entitled "Adolescents Accessing School‐Based vs. Family Planning Clinics: Chlamydia and Gonorrhea Testing and Treatment Out‐comes". I really appreciate your work and implication.

Your manuscript needs a major revision and i have the following comments:

  1. Please explain the novelty of your study. This research bing important changes in the literature?
  2. Please explain us if the patients has a screening CT/GC testing or their has symptoms. In order to calculate the prevalence is mandatory to specify that the CT/GC testing was for screening.
  3. Please revise the references section according instruction for author's.
  4. Knowing this results please specify how we can improve patients care and screening. 
  5. The manuscript required minor punctuation revision.

Reviewer 3 Report

This is an important study to update CT and GC prevalence data. and important to highlight the successes of SBC in reaching this vulnerable population and reducing the barriers to accessing sexual health care. These findings could be a step in the right direction for school-wide population-based screening in Houston/Harris County but I think the paper could be more detailed to explain that connection.

The methods, diagrams, flowchart included are clear and straightforward for the study's process. I think they add a simple visual element to the article and I like it!

The COVID-19 impact section is good - gives a picture of the response and context of SBC & FPCs for the CT and GC data in Houston/Harris County. Could pull this into the future directions on recommendations for future pandemic response and how a partnership with health department, parents, clinical leadership would work for reproductive and sexual health care.

Round 2

Reviewer 2 Report

Dear author’s

Your article has been very well revised and now i

think that is ready for publication.

Author Response

Thank you so much for your time and review. We appreciate it.